# Impacts of Oak Mulch Amendments on Rhizosphere Microbiome of Citrus Trees Grown in Florida Flatwood Soils

**DOI:** 10.3390/microorganisms11112764

**Published:** 2023-11-14

**Authors:** John M. Santiago, Lukas M. Hallman, John-Paul Fox, Marco Pitino, Robert G. Shatters, Liliana M. Cano, Lorenzo Rossi

**Affiliations:** 1Horticultural Sciences Department, Indian River Research and Education Center, Institute of Food and Agricultural Sciences, University of Florida, Fort Pierce, FL 34945, USA; 2Plant Pathology Department, Indian River Research and Education Center, Institute of Food and Agricultural Sciences, University of Florida, Fort Pierce, FL 34945, USA; 3Horticultural Research Laboratory, U.S. Department of Agriculture, Agricultural Research Services, Fort Pierce, FL 34945, USA

**Keywords:** citrus, flatwoods, microbiome, soil amendments, soil nutrient

## Abstract

Rhizosphere interactions are an understudied component of citrus production. This is even more important in Florida flatwood soils, which pose significant challenges in achieving sustainable and effective fruit production due to low natural fertility and organic matter. Citrus growers apply soil amendments, including oak mulch, to ameliorate their soil conditions. Thus, the aim of this research was to evaluate the effects of oak mulch on citrus nutrient uptake, soil characteristics, and rhizosphere composition. The plant material consisted of ‘Valencia’ sweet orange (*Citrus × sinensis*) trees grafted on ‘US-812’ (*C. reticulata × C. trifoliata*) rootstock. The experiment consisted of two treatments, which included trees treated with oak mulch (300 kg of mulch per plot) and a control. The soil and leaf nutrient contents, soil pH, cation exchange capacity, moisture, temperature, and rhizosphere bacterial compositions were examined over the course of one year (spring and fall 2021). During the spring samplings, the citrus trees treated with oak mulch resulted in significantly greater soil Zn and Mn contents, greater soil moisture, and greater rhizosphere bacterial diversity compared to the control, while during the fall samplings, only a greater soil moisture content was observed in the treated trees. The soil Zn and Mn content detected during the spring samplings correlated with the significant increases in the diversity of the rhizosphere bacterial community composition. Similarly, the reduced rates of leaching and evaporation (at the soil surface) of oak mulch applied to Florida sandy soils likely played a large role in the significant increase in moisture and nutrient retention.

## 1. Introduction

Florida flatwood soils used in citrus production are extremely sandy (up to 98%), poorly developed, consist of low soil organic matter (S.O.M.) and are prone to nutrient leaching [1]. Additionally, citrus production in the state has undergone a significant decline since 2006 [2,3,4], much of which has been caused by a disease known as huanglongbing (HLB). HLB-affected trees have been shown to exhibit premature fruit drop, eventually resulting in trees being unable to bear fruit [5,6]. Furthermore, HLB has also been shown to impair the root system of citrus trees, often before foliar symptoms are visible, leading to reduced biomass [7]. The majority of the root biomass loss occurs in the fibrous roots responsible for nutrient uptake [8].

The application of soil amendments may prove useful as a management strategy to mitigate HLB disease symptoms while ameliorating soil qualities. Soil amendments serve as a sustainable option to improve the soil quality and plant physiology of agroecosystems, mainly by influencing the soil characteristics, such as the water-holding capacity, porosity, surface area, and, notably, S.O.M. content [9]. Additional functions of soil amendments also include the stabilization of the soil structure and reduced soil bulk density, facilitating the healthy growth and establishment of root systems [10]. Biochar, for instance, is a soil amendment produced from a process called pyrolysis, and provides high water and nutrient retention [11], reduced need for fertilizers [12], and improved crop yield and fruit quality [13]. Additionally, compost also serves as a commonly applied soil amendment in agricultural production, enhancing both the soil quality and plant health through increases in the cation exchange capacity (C.E.C.), water retention, S.O.M., and beneficial soil microorganisms [14].

Interestingly, use of oak mulch as a soil amendment with citrus has gained some interest, as anecdotal reports from Florida growers suggest that citrus trees growing within the drip line of large oak trees showed minimal HLB symptoms, whereas citrus trees not under the oak tree drip line exhibited severe HLB symptoms [15]. Particularly, a study carried out by Pitino et al. [16] reported that citrus trees treated with oak leaf extracts displayed an increase in physiological parameters and nutrient uptake concurrently with a reduction of *C*Las titer, when compared to citrus trees treated with just water. Additionally, one study from Hallman et al. [17] found that oak mulch applied to 4-year-old ‘Valencia’ sweet orange trees grafted on ‘US-812’ led to greater soil moisture and soil nutrient concentrations of phosphorus (P), potassium (K), and magnesium (Mg) compared to trees grown on non-mulched plots. Mulumba and Lal [18] found that as little as 2000 kg/ha of mulch can have a statistically significant effect on the soil characteristics. Changes in the soil characteristics from oak mulch applications can also have subsequent effects on the soil microorganisms that reside within the rhizosphere of citrus.

The rhizosphere is an area of soil that surrounds the roots of plants and hosts colonies of microorganisms, such as bacteria and fungi, many of which are capable of forming unique plant–host symbiotic relationships [19]. Microorganisms from the bulk soil are recruited to the rhizosphere through the release of root exudates, which vary according to plant species, age, and stress levels [20]. During periods of abiotic stress, greater amounts of root exudates are released, signaling microorganisms to migrate from the bulk soil to the rhizosphere, much of which alleviate stress in a variety of different ways [21]. Bacteria within the rhizosphere that have the capability to promote plant development are known as plant growth-promoting rhizobacteria (PGPR) [22,23]. Microorganisms known as PGPR can assist plant development in a variety of ways, some of which include converting immobile nutrients into a soluble form for root uptake, as well as naturally outcompeting pathogenic organisms that may be detrimental to plant health [24]. Select genera of bacteria have been identified to display PGPR activity, some of which include *Pseudomonas*, *Bacillus*, and *Rhizobium* [23,25]. Species of *Bacillus*, for instance, have been shown to produce plant growth regulators, such as indole-3-acetic acid (IAA), which shorten plant growth cycles [26].

Recently, HLB has been shown to decrease the abundance and diversity of rhizosphere microorganisms, thereby reducing key functional properties and impairing plant host–microbiome interactions [27]. There is still limited information associated with the impacts of soil amendments on the microorganisms of the rhizosphere of citrus trees. Investigating the implications of oak mulch applications on root systems and the root microbiome can better assist in the development of more refined management practices. The objective of this study was to determine the influence of oak mulch applications on the rhizosphere microbiome of sweet oranges trees. It is predicted that oak mulch applied to citrus trees will result in a significantly more diverse rhizosphere composition.

## 2. Materials and Methods

### 2.1. Site Description and Plant Materials

The experiment took place at the United States Department of Agriculture, Agricultural Research Service (USDA-ARS) Picos farm located in Fort Pierce, Florida, USA (27°26′01.2″ N 80°25′51.0″ W). The plant material consisted of 7-year-old ‘Valencia’ sweet orange (*Citrus × sinensis*) trees grafted on ‘US-812’ (*Citrus reticulata × Citrus trifoliata*) rootstock. The trees were grown in soil that was classified as Oldsmar fine sand, a sandy, siliceous, hyperthermic family of Alfic Arenic Haplaquads Glossaqualfs. The experiment consisted of two treatments, which included trees treated with oak mulch (300 kg of mulch per plot) and trees not treated with oak mulch (control). The dimensions of the plots were 2.74 m × 10.97 m. Branches from adult laurel oak trees (*Quercus laurifolia*) were used as a source material for the oak mulch amendments (Figure 1A). The frequency of the oak mulch applications was once per year and took place during September from 2020 to 2022. A total of 24 trees were utilized in this experiment, which were split into 12 trees per treatment (oak mulch-applied and control). Additionally, the trees were separated into 3 plots per treatment, each plot consisting of 4 trees (Figure 1B). The plots were managed with identical irrigation, fertilization, and other management practices.

### 2.2. Leaf Nutrient Analysis

A nutrient analysis was conducted on the leaf samples collected in spring and fall after the first mulch application (April and October 2021). A total of 4 leaves were collected from all 4 trees in each experimental plot, which were later pooled to create 1 compound sample per experimental plot. When selecting the leaf samples for nutrient analysis, only those that were mature and fully expanded were collected.

Deionized water was used to rinse and separate any excess nutrients on the surface of leaf samples, which were later dried at 80 °C. The leaves were then processed through a 1.0 mm mesh screen with a Thomas Wiley mill (Thomas Scientific, Swedesboro, NJ, USA). A 20 mL tube was then used to store the ground leaves at room temperature.

Once ready for analysis, the leaves were analyzed for their boron (B), zinc (Zn), manganese (Mn), and iron (Fe) concentrations using an inductively coupled argon plasma emission (ICP-MS) spectrophotometer (Spectro Ciros CCD, Fitzburg, MA, USA) [28]. The nutrient concentrations were expressed by percentage of dry tissue mass. The leaf nutrient contents that were measured included nitrogen (N), phosphorus (P), potassium (K), magnesium (Mg), calcium (Ca), sulfur (S), boron (B), zinc (Zn), manganese (Mn), iron (Fe), and copper (Cu).

### 2.3. Soil Nutrient, Temperature, and Moisure Analysis

Soil nutrient analyses were conducted on the samples collected at the same time of the leaf nutrient sampling (April and October 2021). A total of 4 soil cores were collected within the dripline of all 4 trees per experimental plot using a soil auger (One-Piece Auger model #400.48, AMS, Inc., American Falls, ID, USA), which was 7 cm in diameter and 10 cm in depth. The cores were later pooled to form one 1 L sample per plot, which were later dried overnight. The nutrient concentrations were determined using Mehlich-3 extraction [29]. Approximately 25 mL of Mehlich III extractant solution (0.2 M CH_3_COOH + 0.015 M NH_4_F + 0.013 M HNO_3_ + 0.001 M EDTA + 0.25 M H_4_NO_3_) was pipetted into an extraction tube containing the dry soil samples of 2.5 g (±0.05). The soil nutrient concentrations were determined using inductively coupled plasma optical emission spectroscopy (ICP-OES, Spectro Ciros CCD, Fitzburg, MA, USA) [30]. The soil nutrient contents that were measured included P, K, Mg, Ca, S, B, Zn, Mn, Fe, and Cu. Additionally, the soil organic matter (S.O.M.), pH, and cation exchange capacity (C.E.C.) were also measured.

A traceable digital pocket soil thermometer (Digital Pocket Thermometer, Traceable, Webster, TX, USA) was used to record 1 soil temperature reading per experimental plot in April and October 2021. One soil temperature reading was taken at each plot every month.

The soil moisture was measured in April and October 2021 using an HH2 Moisture Meter coupled with a Delta-T soil moisture sensor (HH2 Moisture Meter, Delta-T Devices, Cambridge, UK). For each of the 4 trees per plot, 4 soil moisture readings were collected.

### 2.4. Sampling of Rhizosophere Soil and DNA Isolation

The rhizosphere samples were taken directly after bulk soil sampling (April and October 2021), which consisted of the rhizosphere soil (located around the roots) being lightly shaken from the roots and placed in 50 mL sterile tubes. Two rhizosphere samples were collected per experimental plot, providing a total of 6 samples per treatment. Approximately 50 g of rhizosphere soil was collected from each plant. The samples were stored at −20 °C prior to DNA extraction. Approximately 15 mL of sterile phosphate-buffered saline (800 mL distilled water, 8 g NaCl, 0.2 g KCl, 1.44 g Na_2_PO_4_, 0.24 g KH_2_PO_4_) were added to the sample and shaken by hand for 15 s. The roots were removed with forceps and discarded, the remaining soil was centrifuged at 3000× *g* for 15 min, and the supernatant was later discarded. Soil DNA was extracted from 0.25 g of the soil pellet using a DNeasy PowerSoil Kit (Qiagen Inc., Germantown, MD, USA) according to the manufacturer’s instructions.

### 2.5. DNA Quantification and PCR Amplification of Rhizophere Soil Samples

The rhizosphere DNA concentrations were quantified through the use of a Qubit fluorometer (Thermofisher Scientific, Waltham, MA, USA). The CS1_515F (5′-ACACTGACGACATGGTTCTACAGTGYCAGCMGCCGCGGTAA-3′) and CS2_926R (5′-TACGGTAGCAGAGACTTGGTCTCCGYCAATTYMTTTRAGTTT-3′) primers were then used in the PCR amplification of the DNA, specifically, the V4-V5 variable regions’ RNA genes. The Genomics and Microbiome Core Facility (GMCF) at Rush University (Chicago, IL, USA) executed both the library preparation and sequencing of the DNA samples according to the methods found in Santiago et al. [31].

### 2.6. Sequencing Analysis and Taxonomic Assignment of Microorganisms Associated with the Rhizosphere Soil Samples

DADA2 was then used to demultiplex the raw read sequences [32] within the Qiime 2 [33] package, which assisted in removing DNA associated with chimeras, primers, and adapters. The SILVA 128 database was used with a naïve Bayes classifier as a reference for 16S rRNA when establishing taxonomic assignments to amplicon sequence variants (ASVs) [34]. The data were later stored under the accession number PRJNA962263.

### 2.7. Plant and Soil Data Statistical Analysis

Through use of R software 4.3 (RStudio, Boston, MA, USA), a Student’s *t*-test was used to split the main effect means with Tukey’s honestly significant difference post hoc test. The differences were deemed significant if the *p* values were less than or equal to 0.05.

### 2.8. Rhizosphere Composition Statistical Analysis

The data were log transformed prior to the alpha and beta bacterial diversity analyses [35] through use of the Phyloseq v1.24.0 package in R [36]. The total observed ASVs were used in the alpha diversity analyses with the Shannon index. The beta diversity analyses were implemented with a principal coordinate analysis (PCoA) on weighted UniFrac distances, from which a two-way analysis of similarities (ANOSIM) test was performed to determine the significant differences among the treatments. A visualization of the interactions found between the bacterial communities in the sweet orange rhizosphere, soil and root parameters, and soil nutrient concentrations was created through the use of a Canonical correspondence analysis (CCA). Significant differences were determined through an ANOSIM, which performed multivariate comparisons of the groups that were obtained. The beta diversity analyses included PCoA on Bray–Curtis distances in April 2021 and October 2021. A PERMANOVA test was performed to determine the significant differences in the beta diversity between the treatments [37].

## 3. Results

### 3.1. Impact of Oak Mulch on Soil and Leaf Nutrient Concentrations

There were no significant differences observed in the leaf macronutrient and micronutrient concentrations between the citrus trees treated with oak mulch and the control in April 2021 (Figure 2A,B). There were no significant differences observed in the soil macronutrients between the citrus trees treated with oak mulch and the control in April 2021 (Figure 2C). The citrus trees treated with oak mulch had a significantly greater soil micronutrient concentration of Zn and Mn compared to those treated with the control in April 2021 (Figure 2D). Significant differences were not found in the leaf macronutrient and micronutrient concentration between the citrus trees treated with oak mulch and the control in October 2021 (Figure 3A,B). Significant differences were not found in the soil macronutrient and micronutrient concentrations between the citrus trees treated with oak mulch and the control in October 2021 (Figure 3C,D).

### 3.2. Impact of Oak Mulch on Soil Characteristics

There were no significant differences found in the pH, temperature, and C.E.C. of the soils treated with oak mulch in April 2021 (Figure 4A–C). The citrus trees treated with oak mulch resulted in significantly greater soil moisture compared to the trees treated with the control in April 2021 (Figure 4D).

There were no significant differences found in the pH, temperature, and C.E.C. of the soils treated with oak mulch in April 2021 (Figure 5A–C). The citrus trees treated with oak mulch resulted in significantly greater soil moisture compared to the trees treated with the control in April 2021 (Figure 5D). Additionally, the S.O.M. increased over time, as was also reported in Hallman et al. [17].

### 3.3. Impact of Oak Mulch on Rhizosphere Bacterial Communities

The rhizosphere bacterial alpha diversity did vary according to the treatments in April 2021 (Figure 6A) and October 2021 (Figure 7A). The citrus trees treated with oak mulch had a significantly greater bacterial alpha diversity compared to those treated with the control in April 2021 (Figure 6A). In October 2021, greater bacterial diversity was also found in the citrus trees treated with oak mulch compared to the trees treated with the control (Figure 7A).

The rhizosphere bacterial beta diversity was significantly impacted by the treatment in April 2021 (Figure 6B). The citrus trees treated with oak mulch resulted in significantly greater rhizosphere bacterial beta diversity compared to those treated with the control in April 2021 (Figure 6B). Additionally, a total explained variance of 46.34% was found using Bray–Curtis distances when estimating the beta diversity between the treatments in April 2021 (Figure 6B). The citrus trees treated with oak mulch had a greater rhizosphere bacterial beta diversity compared to those treated with the control in October 2021 (Figure 7B). Additionally, the Bray–Curtis distances had a total explained variance of 49.27% when estimating the beta diversity between the treatments in October 2021 (Figure 7B).

Variation was present in the relative abundance of the bacterial taxonomic orders among the treatments in both April 2021 and October 2021. The citrus trees treated with oak mulch had a rhizosphere bacterial community with a significantly greater relative abundance of *Rhizobiales* (*p* < 0.006), *Vicinamibacterales* (*p* < 0.05), *Planctomycetales* (*p* < 0.01), *Reyranellales* (*p* < 0.03), *Caldilineales* (*p* < 0.02), *Entotheonellales* (*p* < 0.01), *Thermoanaerobaculales* (*p* < 0.05), *Flavobacteriales* (*p* < 0.03), *Bdellovibrionales* (*p* < 0.003), *Pseudomonadales* (*p* < 0.006), and *Babeliales* (*p* < 0.02) compared to those treated with the control in April 2021 (Table 1). Conversely, a significantly greater relative abundance of *Gemmatimonadales* (*p* < 0.05), *Pirellulales* (*p* < 0.03), *Nitrospirales* (*p* < 0.05), *Frankiales* (*p* < 0.003), *Haliangiales* (*p* < 0.03), *Latescibacterota* (*p* < 0.003), Nitrosotaleales (*p* < 0.05), *Kapabacteriales* (*p* < 0.04), *Kallotenuales* (*p* < 0.04), and *Azospirillales* (*p* < 0.03) was found in rhizosphere of the citrus trees treated with the control compared to those treated with the oak mulch in April 2021 (Table 1).

The citrus trees treated with oak mulch had a rhizosphere bacterial community with a significantly greater relative abundance of *Rhizobiales* (*p* < 0.05), *Solirubrobacterales* (*p* < 0.02), *Reyranellales* (*p* < 0.05), *Diplorickettsiales* (*p* < 0.02), *Bdellovibrionales* (*p* < 0.05), *Myxococcales* (*p* < 0.04), *Streptosporangiales* (*p* < 0.04), and *Deinococcales* (*p* < 0.01) compared to those treated with the control in October 2021 (Table 2). Conversely, a significantly greater relative abundance of *Burkholderiales* (*p* < 0.05), *Gemmatimonadales* (*p* < 0.006), *Rokubacteriales* (*p* < 0.02), *Nitrospirales* (*p* < 0.02), *Latescibacterota* (*p* < 0.02), *Tepidisphaerales* (*p* < 0.02), *Dadabacteriales* (*p* < 0.05), *Fimbriimonadales* (*p* < 0.05), and *Longimicrobiales* (*p* < 0.05) was found in the rhizosphere of the citrus trees treated with the control compared to those treated with the oak mulch in October (Table 2).

To evaluate the effects of the soil and plant parameters and nutrient concentrations on the rhizosphere bacteria in April and October 2021, CCA was utilized (Figure 8 and Figure 9). Each arrow refers to a parameter, with the length of the arrow corresponding to the relative significance of the interaction shared with the rhizosphere microbiome. An ANOSIM determined that soil P content, moisture, and temperature were significantly correlated with the citrus rhizosphere bacterial communities between the treatments in April 2021 (Figure 8). Additionally, an ANOSIM also determined that soil P content, moisture, temperature, pH, and C.E.C. were significantly correlated with the citrus rhizosphere bacterial communities between the treatments in October 2021 (Figure 9).

## 4. Discussion

This experiment explored the effects of oak mulch on the citrus rhizosphere bacterial community composition. The application of oak mulch had a notable impact on the soil micronutrient concentrations of the citrus trees, as significantly greater soil Zn and Mn contents were observed in April 2021 (Figure 2D). Similarly, when examining the impacts of white pine (*Pinus strobus*) mulch on both the soil quality and blueberry (*Vaccinium angustifolium*) health, a study conducted by Gumbrewicz and Calderwood [38] identified a significant increase in soil Mn and Ca content, as well as S.O.M. and C.E.C. The use of oak mulch as a soil amendment likely caused decreases in evaporation at the soil surface and reduced the rates of leaching, both contributing toward increases soil nutrient retention [39], which may also explain the significant increases in soil moisture observed in both April 2021 (Figure 4D) and October 2021 (Figure 5D).

The relative abundances of the bacterial populations in the rhizosphere were estimated to range anywhere between 10 and 100 times greater than in the bulk soil [37]. The diversity of bacteria within the rhizosphere serves as a useful indicator in determining the health status of the plant-host [21]. Typically, greater bacterial diversity in the rhizosphere correlates with benefits in plant health, specifically through key microbe functions, such as enhanced growth and disease suppression [40]. In this study, the significantly greater soil Zn and Mn content (Figure 2D) found in the oak mulch-treated soils of the citrus trees may be related to significant shifts in the rhizosphere bacterial alpha (Figure 6A) and beta diversity (Figure 6B) observed in April 2021. Interestingly, Wu et al. [41] identified host–pathogen interactions in soils with increased Zn concentrations that suppressed the growth of pathogenic organisms, benefiting the soil quality and plant health (i.e., more diverse rhizosphere community composition). Moreover, a lack of significantly greater Mn and Zn content in the oak mulch versus the control-treated soils in October 2021 may also explain why significant differences were not observed in the citrus tree rhizosphere alpha and beta diversity.

The application of oak mulch to the soil of the citrus trees may have had a significant effect on the relative abundance of rhizosphere bacteria, notably, the PGPR that assist plant health. *Rhizobiales* was not only found among the top 10 most abundant orders of bacteria in the soils from both treatments but was also significantly more abundant in the rhizosphere of the citrus trees treated with oak mulch in April 2021 (Table 1) and October 2021 (Table 2). *Rhizobiales* are soil bacteria that have been shown to establish symbiotic relationships with plant-hosts, assisting in the production of auxins, vitamins, and nitrogen fixation. *Rhizobiales* is associated with host plant benefit nutrient acquisition and plants’ defenses against abiotic and biotic stresses [42]. Significantly, a greater relative abundance of *Vicinamibacterales* in the citrus tree rhizosphere may have been due to the use of oak mulch. *Vicinamibacterales* has been shown to increase the availability of soil N and P content, as they have the ability to solubilize these nutrients into a form that is readily available for root uptake, increasing the available pool of resources to benefit plant health [43].

The CCA suggests that a strong relationship was shared among the measured soil parameters and the rhizosphere bacterial communities among the treatments in April 2021 (Figure 8) and October 2021 (Figure 9). Changes in the soil characteristics (e.g., moisture, temperature, and pH) can have notable impacts on the rhizosphere community structure and composition [44,45]. Notably, the soil moisture significantly correlated with the citrus tree rhizosphere bacterial communities between the oak mulch and control-treated soils in April 2021 (Figure 8) and October 2021 (Figure 9). Similarly, [46] found that increases in the soil moisture resulted in greater microbial diversity of both fungi and bacteria within the rhizosphere of sandy elm (*Ulmus pumila*). Additionally, [47] identified significant declines in the diversity and relative abundance of soil bacteria and fungi when exposed to arid conditions with reduced moisture. Soil moisture can influence soil microbial activity, including S.O.M., in a variety of different ways, some of which include the capability to effect competition of N between plants and soil microorganisms [48], aggregate stability [49], and root exudation [50].

Additionally, the CCA infers that the rhizosphere bacterial communities were also significantly correlated with the soil temperature between the oak mulch and the control-treated soils in April 2021 (Figure 8) and October 2021 (Figure 9). When examining the effects of temperature on the rhizosphere bacterial communities of wild apple trees (*Malus sieversii*), [51] found a positive correlation between the soil temperature and community structure and diversity. Soil temperature is known as a soil-limiting factor that can impact the establishment and respiration of soil microorganisms, enzyme dynamics, and plant development [52], likely playing a role in the observed differences in the rhizosphere community composition between the oak mulch and control-treated soils of the citrus trees (Figure 6B).

Although there were no significant differences in the soil pH between the treatments, a significant correlation between the soil pH and citrus tree rhizosphere bacterial communities among the treatments was also observed with the CCA in October 2021 (Figure 9). Similarly, [53] found that soil pH had a significant effect on the rhizosphere bacterial communities of strawberry, as acidic soils (less than 5.5) resulted in significant microbial diversity compared to neutral soils (7.0–7.5). Soil pH plays an essential role in the interaction and assemblage process of the microbial community in the rhizosphere, as it is directly related to the availability of nutrients [54]. Further analysis into the interaction shared between the conditions of the oak mulch-applied soils, nutrient availability, and rhizosphere community composition may prove useful in optimizing the management strategies for citrus production.

## 5. Conclusions

This study examined the impact of oak mulch on the rhizosphere community composition of HLB-affected citrus trees. The application of oak mulch led to significantly greater soil moisture and soil Zn and Mn content in April 2021, which may be correlated with the significant increase in the diversity of rhizosphere bacterial community composition. The reduced rates of leaching and evaporation (at the soil surface) of oak mulch-applied Florida sandy soils likely played a large role in the significant increase in moisture and nutrient retention. Correlations were identified between the nutrient concentrations, soil moisture and temperature, and rhizosphere bacterial community composition between the treatments. Future experimentation and samplings are required to provide further insight on the interactions shared among the citrus tree rhizosphere bacterial community composition, seasonality, environmental parameters, and oak mulch use.

## Figures and Tables

**Figure 1 microorganisms-11-02764-f001:**
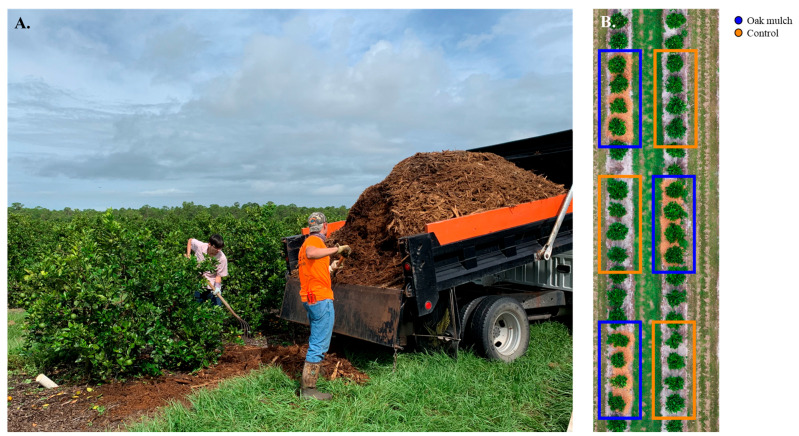
Experimental plots. (**A**) Authors John Santiago and Lukas Hallman applying oak mulch in September 2020 to 7-year-old ‘Valencia’ sweet orange (*Citrus × sinensis*) trees grafted on ‘US-812’ (*Citrus reticulata × Citrus trifoliata*) rootstock. Trees were planted in flatwood soils located in Fort Pierce, FL, USA, and treated with and without oak mulch. (**B**) Aerial view of the experimental design of the field study.

**Figure 2 microorganisms-11-02764-f002:**
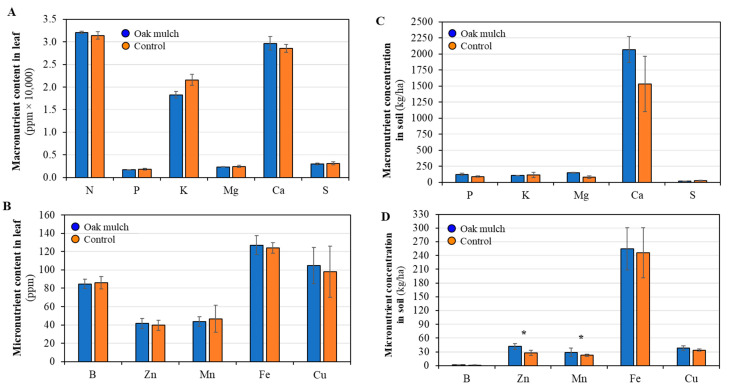
Macro- and micronutrient contents and concentrations measured in April 2021 in leaves (**A**,**B**) and soil (**C**,**D**) of 7-year-old ‘Valencia’ sweet orange (*Citrus × sinensis*) trees grafted on ‘US-812’ (*Citrus reticulata × Citrus trifoliata*) rootstock. Trees were planted in flatwood soils located in Fort Pierce, FL, USA, and treated with and without oak mulch. Graphs (**A**,**C**) are macronutrients; (**B**,**D**) are micronutrients. Bars are ± standard deviation of the mean (*n* = 3). Treatments with * were considered significantly different by Student’s *t*-test with Tukey’s honestly significant difference (*p* < 0.05).

**Figure 3 microorganisms-11-02764-f003:**
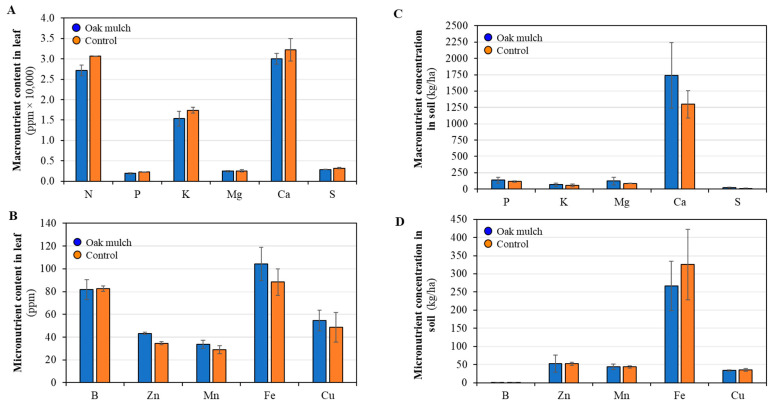
Macro- and micronutrient contents and concentrations measured in October 2021 in leaves (**A**,**B**) and soil (**C**,**D**) of 7-year-old ‘Valencia’ sweet orange (*Citrus × sinensis*) trees grafted on ‘US-812’ (*Citrus reticulata × Citrus trifoliata*) rootstock. Trees were planted in flatwood soils located in Fort Pierce, FL, USA, and treated with and without oak mulch. Graphs (**A**,**C**) are macronutrients; (**B**,**D**) are micronutrients. Bars are ± standard deviation of the mean (*n* = 3).

**Figure 4 microorganisms-11-02764-f004:**
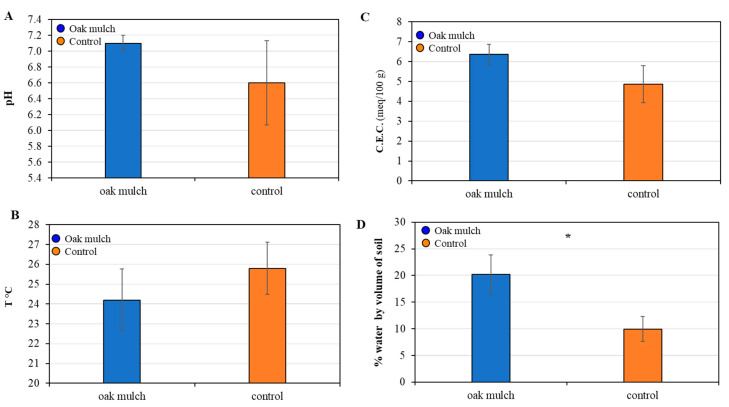
pH (**A**), temperature (**B**), cation exchange capacity (C.E.C.; (**C**)), and moisture (**D**) measured in April 2021 from soils of 7-year-old ‘Valencia’ sweet orange (*Citrus × sinensis*) trees grafted on ‘US-812’ (*Citrus reticulata × Citrus trifoliata*) rootstock. Trees were planted in flatwood soils located in Fort Pierce, FL, USA, and treated with oak mulch and the control. Bars are ± standard deviation of the mean (*n* = 3). Treatments with * were considered significantly different by Student’s *t*-test with Tukey’s honestly significant difference (*p* < 0.05).

**Figure 5 microorganisms-11-02764-f005:**
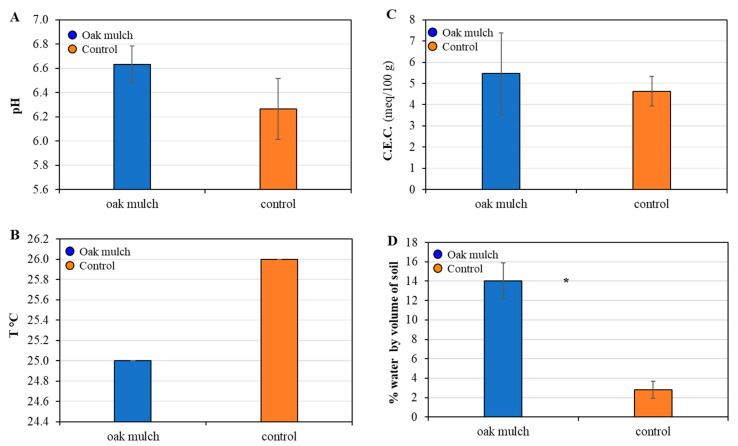
pH (**A**), temperature (**B**), cation exchange capacity (C.E.C.; (**C**)), and moisture (**D**) measured in October 2021 from soils of 7-year-old ‘Valencia’ sweet orange (*Citrus × sinensis*) trees grafted on ‘US-812’ (*Citrus reticulata × Citrus trifoliata*) rootstock. Trees were planted in flatwood soils located in Fort Pierce, FL, USA, and treated with oak mulch and the control. Bars are ± standard deviation of the mean (*n* = 3). Treatments with * were considered significantly different by Student’s *t*-test with Tukey’s honestly significant difference (*p* < 0.05).

**Figure 6 microorganisms-11-02764-f006:**
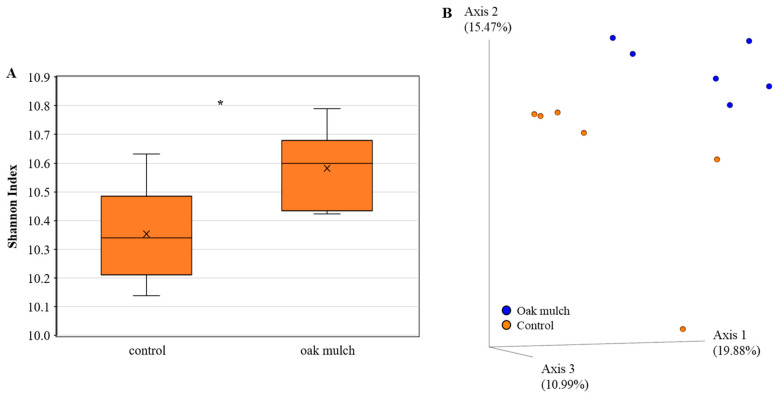
Diversity of rhizosphere bacteria of 7-year-old ‘Valencia’ sweet orange (*Citrus × sinensis*) trees grafted on ‘US-812’ (*Citrus reticulata × Citrus trifoliata*) rootstock from samples taken in April 2021. Trees were planted in flatwood soils located in Fort Pierce, FL, USA, and treated with oak mulch and the control. Alpha diversity (**A**) was measured by Shannon index of rhizosphere bacteria orders among treatments. Plotted in (**A**) are boxes (interquartile), median (line within each box), and whiskers (lowest and greatest values). A principal coordinate analysis (PCoA) based on Bray–Curtis dissimilarity matrix of rhizosphere bacterial samples can be found in (**B**), where colors indicate treatment and include covered (blue) and uncovered (orange). Treatments with * were considered significantly different (*p* < 0.05).

**Figure 7 microorganisms-11-02764-f007:**
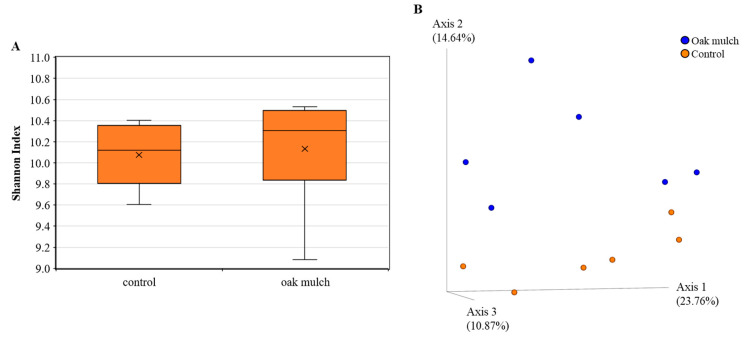
Diversity of rhizosphere bacteria of 7-year-old ‘Valencia’ sweet orange (*Citrus × sinensis*) trees grafted on ‘US-812’ (*Citrus reticulata × Citrus trifoliata*) rootstock from samples taken in October 2021. Trees were planted in flatwood soils located in Fort Pierce, FL, USA, and treated with oak mulch and the control. Alpha diversity (**A**) was measured by Shannon index of rhizosphere bacteria orders among treatments. Plotted in (**A**) are boxes (interquartile), median (line within each box), and whiskers (lowest and greatest values). A principal coordinates analysis (PCoA) based on Bray–Curtis dissimilarity matrix of rhizosphere bacterial samples can be found in (**B**), where colors indicate treatment and include covered (blue) and uncovered (orange).

**Figure 8 microorganisms-11-02764-f008:**
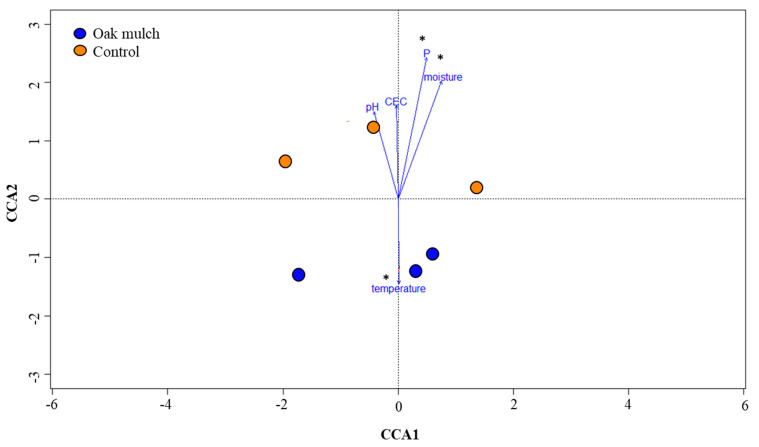
Canonical correspondence analysis (CCA) of rhizosphere bacterial communities of 7-year-old ‘Valencia’ sweet orange (*Citrus × sinensis*) trees grafted on ‘US-812’ (*Citrus reticulata × Citrus trifoliata*) rootstock from samples taken in April 2021. Trees were planted in flatwood soils located in Fort Pierce, FL, USA, and treated with oak mulch and the control. Each dot represents the rhizosphere bacterial community within a sample, while the colors indicate treatment (trees grown with oak mulch in blue and trees grown with the control in orange). Measured parameters include soil phosphorus (P), cation exchange capacity (C.E.C.), soil temperature, soil moisture, and soil pH. Treatments with * were considered significantly different (*p* < 0.05).

**Figure 9 microorganisms-11-02764-f009:**
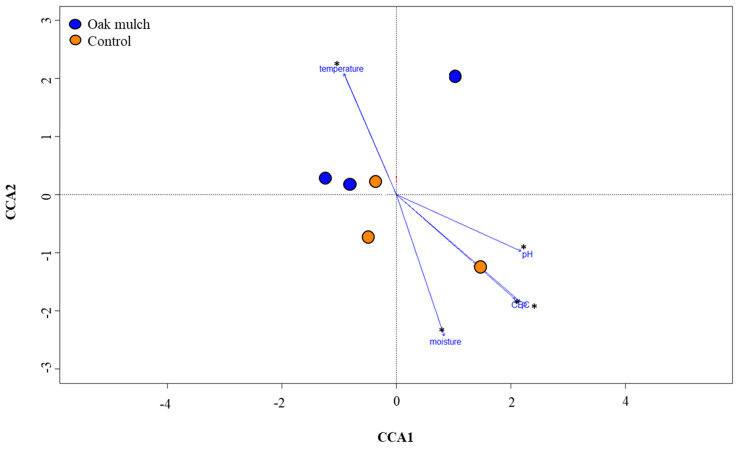
Canonical correspondence analysis (CCA) of rhizosphere bacterial communities of 7-year-old ‘Valencia’ sweet orange (*Citrus × sinensis*) trees grafted on ‘US-812’ (*Citrus reticulata × Citrus trifoliata*) rootstock from samples taken in October 2021. Trees were planted in flatwood soils located in Fort Pierce, FL, USA, and treated with oak mulch and the control. Each dot represents the rhizosphere bacterial community within a sample, while the colors indicate treatment (trees grown with oak mulch in blue and trees grown with the control in orange). Measured parameters include soil phosphorus (P) content, cation exchange capacity (C.E.C.), soil temperature, soil moisture, and soil pH. Treatments with * were considered significantly different (*p* < 0.05).

**Table 1 microorganisms-11-02764-t001:** Relative abundance of bacterial orders in the rhizosphere of 7-year-old ‘Valencia’ sweet orange (*Citrus × sinensis*) trees grafted on ‘US-812’ (*Citrus reticulata × Citrus trifoliata*) rootstock from April 2021. Trees were planted in flatwood soils located in Fort Pierce, FL, USA, and treated with oak mulch and the control.

Bacteria	Oak Mulch	Control	*p*-Value
*Rhizobiales*	15.78%	11.20%	0.006
*Gemmatimonadales*	2.26%	4.18%	0.05
*Vicinamibacterales*	5.43%	4.01%	0.05
*Pirellulales*	1.13%	2.82%	0.03
*Nitrospirales*	1.13%	2.14%	0.05
*Frankiales*	0.34%	1.12%	0.003
*Haliangiales*	0.67%	1.00%	0.03
*Planctomycetales*	1.60%	0.97%	0.01
*Latescibacterota*	0.29%	0.75%	0.003
*Reyranellales*	0.73%	0.32%	0.03
*Nitrosotaleales*	0.03%	0.32%	0.05
*Caldilineales*	0.35%	0.19%	0.02
*Entotheonellales*	0.34%	0.19%	0.01
*Thermoanaerobaculales*	0.31%	0.17%	0.05
*Flavobacteriales*	0.27%	0.14%	0.03
*Bdellovibrionales*	0.24%	0.12%	0.003
*Kapabacteriales*	0.05%	0.10%	0.04
*Kallotenuales*	0.03%	0.09%	0.04
*Pseudomonadales*	0.26%	0.07%	0.006
*Azospirillales*	0.01%	0.04%	0.03
*Babeliales*	0.32%	0.02%	0.02

**Table 2 microorganisms-11-02764-t002:** Relative abundance of bacterial orders in the rhizosphere of 7-year-old ‘Valencia’ sweet orange (*Citrus × sinensis*) trees grafted on ‘US-812’ (*Citrus reticulata × Citrus trifoliata*) rootstock from October 2021. Trees were planted in flatwood soils located in Fort Pierce, FL, USA, and treated with oak mulch and the control.

Bacteria	Oak Mulch	Control	*p*-Value
*Rhizobiales*	14.28%	10.65%	0.05
*Burkholderiales*	5.74%	7.36%	0.05
*Gemmatimonadales*	2.28%	4.70%	0.006
*Rokubacteriales*	1.28%	3.04%	0.02
*Nitrospirales*	1.15%	2.38%	0.02
*Solirubrobacterales*	3.48%	2.25%	0.02
*Latescibacterota*	0.23%	0.94%	0.02
*Tepidisphaerales*	0.06%	0.34%	0.02
*Reyranellales*	0.67%	0.34%	0.05
*Diplorickettsiales*	0.48%	0.23%	0.02
*Dadabacteriales*	0.07%	0.22%	0.05
*Fimbriimonadales*	0.07%	0.19%	0.05
*Bdellovibrionales*	0.28%	0.16%	0.05
*Myxococcales*	0.28%	0.17%	0.04
*Kapabacteriales*	0.06%	0.13%	0.05
*Longimicrobiales*	0.02%	0.07%	0.05
*Streptosporangiales*	0.15%	0.06%	0.04
*Deinococcales*	0.04%	0.005%	0.01

## Data Availability

The datasets generated during and/or analyzed during the current study have been deposited to NCBI. The BioProject accession number is PRJNA962263.

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
