# Peer review of "Impacts of Oak Mulch Amendments on Rhizosphere Microbiome of Citrus Trees Grown in Florida Flatwood Soils"

_microorganisms, 2023, doi:10.3390/microorganisms11112764_

Round 1

Reviewer 1 Report

Comments and Suggestions for Authors

The Abstract section should be reorganized. For example, the contents from Line 14- line 27 are too long and should be greatly shortened, while more main results should be added in the Abstract. There is only one sentence mentions the main results. The conclusion is lacking in the Abstract section.

Most of the figures can be prepared in a more readable typeface.

The authors mentioned many contents relating to huanglongbing. However, there is no disease data in the results.

The authors mentioned soil health, but there are no data of the pathogen. The authors mentioned the higher alpha diversity represent soil health, but this not validated. 

Author Response

Thank you for taking the time for quickly and thoroughly reviewing our manuscript. Our replies to your comments can be found below.

The Abstract section should be reorganized. For example, the contents from Line 14- line 27 are too long and should be greatly shortened, while more main results should be added in the Abstract. There is only one sentence mentions the main results. The conclusion is lacking in the Abstract section.

The abstract has been revised according to your recommendations.

Most of the figures can be prepared in a more readable typeface.

Figures have been updated to improve readability.

The authors mentioned many contents relating to huanglongbing. However, there is no disease data in the results.

Information on huanglongbing was added to introduce the challenges plaguing the Florida citrus industry, and how the disease impacts the root system – and therefore the rhizosphere – of the trees. Disease data was not collected because there is nearly 100% disease incidence in the Indian River growing region, which our research took place in; moreover, since the mulch applications were not meant to influence the disease itself, it was deemed extraneous.

However, with your comments in mind, we trimmed down the amount of background given on HLB to limit it only to the vital information important to this study.

The authors mentioned soil health, but there are no data of the pathogen. The authors mentioned the higher alpha diversity represent soil health, but this not validated.

Rhizosphere microorganism diversity is primarily affected by the bulk soil; therefore, healthy soils promote diversity within the Rhizosphere microbiome. Here is a citation. https://doi.org/10.1007/s11104-015-2446-0

Therefore, rhizosphere microorganism diversity can be used as an indicator of soil health. https://www.ars.usda.gov/research/publications/publication/?seqNo115=144331

However, we realize that “soil health” can be a confusing term so we reworded the manuscript focusing more on the influence of oak mulch applications on the rhizosphere microbiome of sweet oranges trees.

Reviewer 2 Report

Comments and Suggestions for Authors

Abstract: I think it is not necessary to give the exact names of the isolation sets or the method of bioinformatics data processing in the abstract. L24-27. instead, the most important results should be highlighted.

L57-68: are there any statistics indicating how much of this oak mulch is used? Is it really a significant treatment in the context of crops?

L94: is the mere wording that the diversity will be higher relevant? Or should we rather focus on the microorganisms that have specific functions in this rhizosphere, that are promoters?

L97-108: I think a photo of the plots would be a very interesting addition.

L103: plot - please give the dimensions of such a plot.

L104: what was the control?

L102-106: were there any other treatments applied to these plots? Irrigation, fertilisation etc.?

L111: please list what specific nutrients were determined.

L132: how much soil by weight was obtained?

L145-148: thank you very much for such a thorough description and honest approach to the subject of the rhizosphere. There is a great deal of work in the literature where researchers refer to soil taken with a probe "next to" the roots as the rhizosphere. I very much appreciate your approach!

L126: please list which specific nutrients were determined. Why were C, Corg., humus, SOM not determined? Carbon is a key component of the soil environment... This is a big lack. As is a more accurate analysis of N content. Nitrogen exists in many forms in soil.

L160: please provide the sequences of the primers and the literature on which these were chosen.

L174: why was version 3.6.0. used when the stable version 4.3. is already in use? Version 3.6. is from 2019.

Figure 1 and 2: how was nutrient content converted to kg/ha? The caption for the X-axis is missing. I would also suggest to present the values from figure A and B in the same unit. And use the same notation for figures, numbers in all figures, i.e. the same number of decimal places in the numerical notation. The Y-axis caption should read "macro- micronutrient content/concentration in soil/leaf".

L209: what statistical test was used to determine differences? at what n?

L218: what is CEC? There is no abbreviation for this in the methodology.

Figures: perhaps it would be useful to introduce some abbreviation for samples treated with oak mulch?

Figures 3 and 4: X-axis signature is missing. Use the same notation for figures, numbers on all figures, i.e. same number of decimal places in the numerical notation.

L230; 237: what statistical test was used to determine differences? at what n?

L245: this is the methodology. Should be in the description of the methodology. Also Bray-Curtis.

L256-and throughout the paper: the latest 2023 International Committee on Systematics of Prokaryotes (ICSP) guidelines indicate italicising all taxonomic names of microorganisms. Please correct this throughout the paper.

L258 onwards: 'relative abundance'. Use the full expression anywhere in the manuscript.

Figure 5b and 6b: very poor quality of the figure, overlapping captions on the axes. Please increase the font in the figures in the first place.

Figure 5a and 6a: the caption for the X-axis is missing. Please use the same notation for figures, numbers on all figures, i.e. the same number of decimal places in the numerical notation. Please change the Y-axis caption - this is simply the Shanonn index. Please specify in the caption what taxonomic level was used to calculate the index.

Table 1: please adjust the width of column 1 so that it does not break and move names to subsequent rows. "Frequency?" Probably "abundance".

Figures 7 and 8: Poor figure quality, overlapping line captions, illegible axis captions (figures). Please, above all, increase the font in the figures.

I miss a comparison of the results in terms of dates, i.e. a juxtaposition of the results from April 2021 with October 2021. It would be tempting to do such a comparison and see if the parameters have changed during the season.

In addition, I recommend adding climate data in the description of the experiment or supplement. air temperature, precipitation. From the year in which the research was conducted.

Why were results presented only for orders? How were the differences arranged at the genera level?

L350,408 and earlier throughout the text: not "soil Zn", but soil Zn content - and this applies to the entire paper. Please do not use such mental shortcuts.

Comments on the Quality of English Language

Apart from the use of mental abbreviations e.g. 'soil Zn', I have no comments on the language.

Author Response

Thank you for your time and your detailed review of our work. Our replies to your comments can be found below.

Abstract: I think it is not necessary to give the exact names of the isolation sets or the method of bioinformatics data processing in the abstract. L24-27. instead, the most important results should be highlighted.

The abstract has been revised according to your recommendations.

L57-68: are there any statistics indicating how much of this oak mulch is used? Is it really a significant treatment in the context of crops?

Other scientific experiments show that as little as 2000 kg/ha of mulch can have a statistically significant effect on soil characteristics. We applied 300 kg/plot which is equal to ~99,666 kg/ha. This citation was added to the paper. https://doi.org/10.1016/j.still.2007.10.011. Additionally, growers in Florida are also implementing this practice in their operations.

L94: is the mere wording that the diversity will be higher relevant? Or should we rather focus on the microorganisms that have specific functions in this rhizosphere, that are promoters?

In our case, focusing on the parameter of diversity better allows us to compare differences among the control and oak mulch application, as we did use microbial diversity as an indicator of plant health. Greater diversity typically allows for a greater incidence of different plant growth promoters to become established in the rhizosphere.

L97-108: I think a photo of the plots would be a very interesting addition.

A photo of the plots was added.

L103: plot - please give the dimensions of such a plot.

The dimensions of the plot are 2.74 m x 10.97 m, and this information was added to the paper.

L104: what was the control?

Added “trees not treated with oak mulch”.

L102-106: were there any other treatments applied to these plots? Irrigation, fertilisation etc.?

Added “Plots were managed with the identical irrigation, fertilization, and other management practices.”.

L111: please list what specific nutrients were determined.

This information was added.

L132: how much soil by weight was obtained?

Soil samples were measured by volume, and “1 L” was added.

L145-148: thank you very much for such a thorough description and honest approach to the subject of the rhizosphere. There is a great deal of work in the literature where researchers refer to soil taken with a probe "next to" the roots as the rhizosphere. I very much appreciate your approach!

Your kind words are so appreciated. I also am thankful for the knowledge and direction that you have given us with this journal article.

L126: please list which specific nutrients were determined. Why were C, Corg., humus, SOM not determined? Carbon is a key component of the soil environment... This is a big lack. As is a more accurate analysis of N content. Nitrogen exists in many forms in soil.

A list of specifically measured nutrients was added, and information about measurements of SOM were added. A longer study that primarily focused on soil characteristics and oak mulch applications can be found here: https://www.frontiersin.org/articles/10.3389/fsoil.2023.1200847/full

L160: please provide the sequences of the primers and the literature on which these were chosen.

Sequences of the primers were added.

L174: why was version 3.6.0. used when the stable version 4.3. is already in use? Version 3.6. is from 2019.

We confirmed that the 4.3 version was used for our statistical analysis, this was a typo, and the error has been changed in the document.

Figure 1 and 2: how was nutrient content converted to kg/ha? The caption for the X-axis is missing. I would also suggest to present the values from figure A and B in the same unit. And use the same notation for figures, numbers in all figures, i.e. the same number of decimal places in the numerical notation. The Y-axis caption should read "macro- micronutrient content/concentration in soil/leaf".

Data in Figures 1 and 2 were first obtained in grams per kilogram and then multiplied by the bulk density of soil resulting in g per cubic meters, then g is converted to kg and cubic meters is converted to ha. Figures have been improved following your comments.

L209: what statistical test was used to determine differences? at what n?

This information has been added to the figure captions.

L218: what is CEC? There is no abbreviation for this in the methodology.

CEC was now added to the methodology.

Figures: perhaps it would be useful to introduce some abbreviation for samples treated with oak mulch?

Thanks for your suggestions. Since we only have 2 treatments we preferred to spell out all the names, so we avoided confusions.

Figures 3 and 4: X-axis signature is missing. Use the same notation for figures, numbers on all figures, i.e. same number of decimal places in the numerical notation.

Figures 3 and 4 have been improved following your comments.

L230; 237: what statistical test was used to determine differences? at what n?

This information was in the materials and methods section and has now been added to the figure captions.

L245: this is the methodology. Should be in the description of the methodology. Also Bray-Curtis.

These lines were shifted to the methodology section.

L256-and throughout the paper: the latest 2023 International Committee on Systematics of Prokaryotes (ICSP) guidelines indicate italicising all taxonomic names of microorganisms. Please correct this throughout the paper.

All taxonomic names were italicized throughout the paper.

L258 onwards: 'relative abundance'. Use the full expression anywhere in the manuscript.

Added relative to all uses of the term “abundance”.

Figure 5b and 6b: very poor quality of the figure, overlapping captions on the axes. Please increase the font in the figures in the first place.

Figures have been improved following your comments.

Figure 5a and 6a: the caption for the X-axis is missing. Please use the same notation for figures, numbers on all figures, i.e. the same number of decimal places in the numerical notation. Please change the Y-axis caption - this is simply the Shanonn index. Please specify in the caption what taxonomic level was used to calculate the index.

Figure captions have been improved following your comments.

Table 1: please adjust the width of column 1 so that it does not break and move names to subsequent rows. "Frequency?" Probably "abundance".

Table 1 has been improved following your comments, and frequency was changed to abundance.

Figures 7 and 8: Poor figure quality, overlapping line captions, illegible axis captions (figures). Please, above all, increase the font in the figures.

Figures have been improved following your comments.

I miss a comparison of the results in terms of dates, i.e. a juxtaposition of the results from April 2021 with October 2021. It would be tempting to do such a comparison and see if the parameters have changed during the season.

Thanks for your suggestions, unfortunately we don’t have that capability at this time since the grant supporting this research has been closed and the sheer quantity of data requires resources to access computing hardware.

In addition, I recommend adding climate data in the description of the experiment or supplement. air temperature, precipitation. From the year in which the research was conducted.

Thank you for your recommendation. Weather data for the area is available at the following link: https://fawn.ifas.ufl.edu/ We tried to find correlations between rain and temperature but did not find any trend, this is why we did not add them to our paper.

Why were results presented only for orders? How were the differences arranged at the genera level?

Data associated with microorganisms with respect to our study was limited in the genus level. Unfortunately, this resulted in us having to utilize microbial data on the taxonomic level of order.

L350,408 and earlier throughout the text: not "soil Zn", but soil Zn content - and this applies to the entire paper. Please do not use such mental shortcuts.

This was fixed throughout the paper.

Round 2

Reviewer 1 Report

Comments and Suggestions for Authors

In the text, the authors used 'soil health', but there is no data directly related to this. Please do not use this item.

In the discussion section, the authors highlighted the role/function of specific bacterial taxa. These statements should be further validated as in previous studies, such as https://doi.org/10.1186/s40168-019-0775-6, https://doi.org/10.1016/j.molp.2023.03.009. This can be further discussed.

For most of the bar plots, the bars are too wide, please revise.